# Robotic Living Donor Right Hepatectomy: A Systematic Review and Meta-Analysis

**DOI:** 10.3390/jcm11092603

**Published:** 2022-05-05

**Authors:** Eddy P. Lincango Naranjo, Estefany Garces-Delgado, Timo Siepmann, Lutz Mirow, Paola Solis-Pazmino, Harold Alexander-Leon, Gabriela Restrepo-Rodas, Rafael Mancero-Montalvo, Cristina J. Ponce, Ramiro Cadena-Semanate, Ronnal Vargas-Cordova, Glenda Herrera-Cevallos, Sebastian Vallejo, Carolina Liu-Sanchez, Larry J. Prokop, Ioannis A. Ziogas, Michail G. Vailas, Alfredo D. Guerron, Brendan C. Visser, Oscar J. Ponce, Andrew S. Barbas, Dimitrios Moris

**Affiliations:** 1Knowledge and Evaluation Research Unit, Mayo Clinic, Rochester, MN 55905, USA; eddypa95@gmail.com (E.P.L.N.); sebastianvallejomd@gmail.com (S.V.); ponceoscarj@gmail.com (O.J.P.); 2Department of Teaching and Research, Hospital Vozandes Quito, Quito 170521, Ecuador; 3Equipo de Investigación de la Sociedad Ecuatoriana de Cirugía Bariátrica y Metabólica (SECBAMET), Quito 170508, Ecuador; estufy28@gmail.com (E.G.-D.); alexanderleonharold@gmail.com (H.A.-L.); gabyrestrepo.grr@hotmail.com (G.R.-R.); manceromontalvo@gmail.com (R.M.-M.); cjponce8@hotmail.com (C.J.P.); ramirocadena93@gmail.com (R.C.-S.); ronnalvargas@gmail.com (R.V.-C.); herreraglenda@yahoo.com (G.H.-C.); 4Division of Health Care Sciences, Center for Clinical Research and Management Education, Dresden International University, 01067 Dresden, Germany; timo.siepmann@ukdd.de; 5Medical School, Universidad Internacional del Ecuador, Quito 170411, Ecuador; 6Department of Neurology, University Hospital Carl Gustav Carus, Technische Universität Dresden, 01307 Dresden, Germany; 7Department of General and Visceral Surgery, Medical Campus Chemnitz of the TU Dresden, 01307 Dresden, Germany; l.mirow@skc.de; 8Department of Otolaryngology-Head and Neck Surgery, School of Medicine, Stanford University, Stanford, CA 94305, USA; paosolpaz18@gmail.com; 9Medical School, Universidad de las Américas, Quito 170503, Ecuador; 10Division of Metabolic and Weight Loss Surgery, Hospital General San Francisco IESS, Quito 170111, Ecuador; 11Division of Metabolic and Weight Loss Surgery, Hospital Metropolitano, Quito 170521, Ecuador; 12Instituto de Medicina Tropical Alexander von Humboldt, Universidad Peruana Cayetano Heredia, Lima 15102, Peru; carolina.liu.s@upch.pe; 13Mayo Clinic Libraries, Mayo Clinic, Rochester, MN 55905, USA; prokop.larry@mayo.edu; 14Division of Hepatobiliary Surgery and Liver Transplantation, Department of Surgery, Vanderbilt University Medical Center, Nashville, TN 37232, USA; ioannis.a.ziogas@vumc.org; 151st Department of Surgery, Laikon General Hospital, National and Kapodistrian University of Athens, 11528 Athens, Greece; mike_vailas@yahoo.com; 16Division of Metabolic and Weight Loss Surgery, Department of Surgery, Duke University, Durham, NC 27705, USA; daniel.guerron@duke.edu; 17Division of Hepatobiliary and Pancreatic Surgery, Department of Surgery, Stanford University, Stanford, CA 94305, USA; bvisser@stanford.edu; 18Frimley Park Hospital, Frimley Health NHS Foundation Trust, Surrey GU16 7UJ, UK; 19Department of Surgery, Duke University, Durham, NC 27705, USA; andrew.barbas@duke.edu

**Keywords:** liver transplantation, robot, systematic review, meta-analysis

## Abstract

**Simple Summary:**

Liver transplantation is the mainstay of treatment for patients with end-stage liver disease or certain types of liver cancer. However, the current organ supply cannot meet the continuously increased number of patients added to the liver transplant waitlist, and thus living donation has been proposed as an alternative to expand the donor pool. Robotic living donor right hepatectomy for adult liver transplantation has shown potential for lower morbidity and better donor outcomes, which can help increase donation and the organ supply. The current systematic review summarizes the available evidence comparing the outcomes of robotic, laparoscopic, and open living donor right hepatectomy.

**Abstract:**

The introduction of robotics in living donor liver transplantation has been revolutionary. We aimed to examine the safety of robotic living donor right hepatectomy (RLDRH) compared to open (ODRH) and laparoscopic (LADRH) approaches. A systematic review was carried out in Medline and six additional databases following PRISMA guidelines. Data on morbidity, postoperative liver function, and pain in donors and recipients were extracted from studies comparing RLDRH, ODRH, and LADRH published up to September 2020; PROSPERO (CRD42020214313). Dichotomous variables were pooled as risk ratios and continuous variables as weighted mean differences. Four studies with a total of 517 patients were included. In living donors, the postoperative total bilirubin level (MD: −0.7 95%CI −1.0, −0.4), length of hospital stay (MD: −0.8 95%CI −1.4, −0.3), Clavien–Dindo complications I–II (RR: 0.5 95%CI 0.2, 0.9), and pain score at day > 3 (MD: −0.6 95%CI −1.6, 0.4) were lower following RLDRH compared to ODRH. Furthermore, the pain score at day > 3 (MD: −0.4 95%CI −0.8, −0.09) was lower after RLDRH when compared to LADRH. In recipients, the postoperative AST level was lower (MD: −0.5 95%CI −0.9, −0.1) following RLDRH compared to ODRH. Moreover, the length of stay (MD: −6.4 95%CI −11.3, −1.5) was lower after RLDRH when compared to LADRH. In summary, we identified low- to unclear-quality evidence that RLDRH seems to be safe and feasible for adult living donor liver transplantation compared to the conventional approaches. No postoperative deaths were reported.

## 1. Introduction

Liver transplantation remains the mainstay of treatment in patients with end-stage liver disease [1]. However, the organ shortage makes transplantation a restricted procedure. This has put pressure on transplant programs to increase the donor pool. Apart from expanding the criteria of eligibility for organ donation from deceased donors, there are solid scientific data supporting the role of living liver donation as a potential solution [2]. Unfortunately, the risks associated with transplantation procedures have dampened the enthusiasm for the wide implementation of living donor transplantation (LDLT) [3].

The first ever reported human living donor liver transplants were first attempted in pediatric populations using open approaches. These procedures had a high rate of complications [4]. Therefore, transplant surgeons tried to overcome the associated risks of LDLT adopting minimally invasive techniques in benign and malignant cases [5]. As such, the first laparoscopic donor hepatectomy was performed in 2002 [6]. In recent years, this technology has become widely adopted in adult-to-adult liver transplantation [7]. However, its complication rates remain high, ranging between 16% and 34% [8]. Advances in hepatobiliary surgery have recently allowed the introduction of robotic approaches as a possible alternative to decrease morbidity and facilitate postoperative recovery [9]. Compared to the conventional laparoscopic approach, robotic surgery overcomes the restricted range of motion and physiological tremor [10].

To date, several studies have investigated the efficacy of robots in living donor transplantation; however, their safety and advantages remain controversial. In this study, we summarize and examine the safety of robotic living donor right hepatectomy (RLDRH) compared to open living donor right hepatectomy (ODRH) or laparoscopic living donor right hepatectomy (LADRH) approaches in donor and recipient patients.

## 2. Materials and Methods

This work was presented at the American Transplant Congress 2021, Laurel, NJ, USA [11], and at the Clinical Congress of the American College of Surgeons 2021, Washington, DC, USA [12]. The review protocol was registered at PROSPERO (CRD42020214313) [13], and this manuscript was written according to the Preferred Reporting Items for Systematic Reviews and Meta-Analyses (PRISMA) guidelines [14].

### 2.1. Eligibility Criteria

Observational or experimental comparative studies assessing the safety of RLDRH in patients older than 18 years who require transplantation, regardless of the underlying liver disease, compared to ODRH or LADRH, were included. Patients with normal anatomy or multiple bile ducts and portal trifurcation, remnant liver more than 30%, and a graft-to-recipient weight ratio over 0.8% were included. Eligible studies needed to report safety outcomes such as postoperative pain, liver function, any type of complications, and mortality. Articles that did not meet one of these criteria were excluded.

### 2.2. Data Sources and Searches

We applied a search strategy developed in collaboration with an experienced librarian (LP) in MEDLINE, EMBASE, Cochrane Central Register of Controlled Trials, Cochrane Database of Systematic Reviews, and Scopus. The search was performed from each database’s inception until September 2020 by using controlled vocabulary, supplemented with keywords related to robotic living donor right hepatectomy in human patients. Conference abstracts were included, and case reports, letters, and literature reviews were excluded. Reference lists of selected studies were searched to identify additional publications. We used Google Translate for studies written in languages different than English, Spanish, and Greek. Our full search strategy is available in the Supplemental Digital Content—Search strategies.

### 2.3. Study Selection

A systematic review software program, DistillerSR [15], was used to upload search results. Articles were reviewed (E.P.L.N., E.G.-D., G.R.-R., R.M.-M., H.A.-L., C.J.P., R.C.-S., S.V., C.L.-S., P.S.-P.) in duplicate and independently. First, titles and abstract of articles were screened, and when one reviewer considered them eligible, they proceeded to full-text screening. In the next phase, full-text articles were assessed for eligibility, and only those included by a pair of reviewers were deemed eligible for this systematic review. Before initiating the screening of abstracts or titles and full-text screening, 10 articles were piloted to harmonize the understanding of the eligibility criteria. Disagreements were resolved by consensus between 3 reviewers (E.P.L.N., E.G.-D., D.M.). Full-text screening agreement was calculated using Cohen’s kappa and it was substantial (k = 0.80).

### 2.4. Data Collection and Management

For all included articles, reviewers (G.R.-R., R.M.-M., H.A.-L., C.J.P., R.C.-S., S.V., C.L.-S., P.S.-P., R.V.-C.) extracted data into a predetermined spreadsheet form created by E.P.L.N. Furthermore, E.P.L.N. and E.G.-D. checked all the extracted data for accuracy and completeness. The following information was extracted: (1) general characteristics of the studies (author, date of publication, country, study design, and data collection period), (2) setting (single center and multicenter), (3) donor hepatectomy surgical approach (robotic, laparoscopic, and open), (4) patient inclusion and selection criteria for the robotic approach, (5) donor preoperative characteristics (age, sex, body mass index, total liver volume, graft volume), (6) donor postoperative outcomes (operative time, estimated blood loss, conversion rate, peak aspartate aminotransferase (AST), alanine aminotransferase (ALT), and total bilirubin, pain score, length of hospital stay (LOS), complications, and mortality), (7) recipient preoperative characteristics (age, sex, body mass index, indications for transplant, and Model for End-Stage Liver Disease (MELD) score), and (8) recipient postoperative outcomes (complications and mortality). Postoperative complications were evaluated using the Clavien–Dindo scale [16] and were also listed individually.

### 2.5. Author Contact

We emailed the first and corresponding authors of the included studies and excluded articles due to wrong outcomes, to request further information. We asked about complications, mortality, liver function tests, and pain scores when these were unclear in the manuscript, or not reported. A second e-mail was sent if authors failed to respond in a 1-week period. A final reminder was sent if authors still did not respond in another 1-week period. Only one author replied and shared data in regard to liver function tests, postoperative pain scores, and the estimated blood loss of robotic recipient patients [17]. These data were not used in the meta-analysis as the author only shared data from one group.

### 2.6. Risk of Bias in Individual Studies

Reviewers (G.R.-R., R.M.-M., H.A.-L., C.J.P., R.C.-S., S.V., C.L.-S., P.S.-P., R.V.-C.) worked in duplicate and independently to assess the risk of bias. Disagreements were resolved by consensus with E.P.L.N. The CLARITY tool [18] was used to assess the risk of bias of cohort studies. The domains evaluated were: (1) whether the selection of exposed and non-exposed cohorts was drawn from the same population; (2) confidence in the assessment of exposure; (3) if our outcome of interest was not present at the start of the study; (4) if the study matched exposed and unexposed for all variables that are associated with the outcome of interest or if the statistical analyses adjusted for these prognostic variables; (5) confidence in the assessment of the presence or absence of prognostic factors; (6) confidence in the assessment of the outcome; (7) if the follow-up of the cohort was adequate; and (8) if co-interventions between groups were similar. CLARITY had four possible responses for each domain: “definitively yes”, “probably yes”, “probably no”, and “definitively no”. We grouped these responses as low, unclear, and high risk of bias. The response “definitively yes” was interpreted as low risk of bias, “probably yes” and “probably no” as unclear risk of bias, and “definitively no” as high risk of bias. The overall score for each included study was calculated with the following criteria: studies with at least one domain considered as “high risk of bias” or with more than three as “unclear risk of bias” were judged to be at an overall high risk of bias; studies with at least two domains at “unclear risk of bias” and without domains assessed as “high risk of bias” were considered to be at an overall unclear risk of bias; and those studies with domains classified as “low risk of bias” without any “unclear” or “high risk of bias” domains were considered at an overall low risk of bias [19].

### 2.7. Statistical Analyses

Relative risks (RRs) and their 95% confidence intervals (CIs) were calculated for dichotomous outcomes by using an intention-to-treat analysis approach. For continuous outcomes, we used the data reported after the intervention and calculated overall mean differences (MDs) with their corresponding 95%CIs. All effect sizes were combined through a random effect model with the restricted maximum-likelihood estimator (REML).

Heterogeneity was assessed using between-study variance estimates (tau squared). The proportion of variability in effect size estimates attributed to between-study heterogeneity was assessed with the *I*^2^ statistic. The statistical program R Studio [20] for R software [21] was employed to perform the analyses and generate forest plots. Medians were converted to means and ranges or interquartile ranges to standard deviations (SDs) [22]. Means and SDs of each variable were pooled using the weighted mean and weighted SD [23,24].

### 2.8. Online Resource

To improve the transparency of our article, all documents used in the screening phases, tables, figures, and analysis codes are freely available at Github: https://github.com/ponceoscarj/Hepatectomy/blob/main/hepatectomy.md (accessed on 23 March 2022).

## 3. Results

### 3.1. Overall Characteristics

After removing duplicates, the search strategy retrieved 413 references. Four comparative studies were included in this study (Figure 1) [17,25,26,27]. One study was prospective [25] and the others retrospective [17,26,27]. A total of 1034 patients underwent surgery: 517 were donors and 517 were recipients. All patients were followed at least for 1 month after the surgery. The studies were conducted in South Korea [17] (*n* = 1), Saudi Arabia [27] (*n* = 1), India [25] (*n* = 1), and China [26] (*n* = 1) (Table 1). Overall, the risk of bias was high in two cohort studies [17,27] and unclear in the other two (Appendix A) [25,26].

### 3.2. Studies Reporting Outcomes of Donors

A total of 151 donors underwent RLDRH (mean age: 32.0 SD ± 4.6; female: *n* = 82 [54.3%]), 248 ODRH (mean age: 34.7 SD ± 5.9; female: *n* = 123 [49.6%]), and 118 LADRH (mean age: 36.9 SD ± 12.1; female: *n* = 44 [37.3%]) (Appendix A). Patients’ inclusion criteria for the robotic approach are detailed in Appendix A. Only one study reported patients’ selection criteria and stated that candidates were at will to choose the robotic procedure after passing the inclusion criteria (Appendix A). The mean total liver volume (mL) in patients that underwent RLDRH was lower than the liver volume of ODRH and LADRH patients (1172.0 SD ± 18.4; 1201.5 SD ± 27.6; 1253.4 SD ± 216.2, respectively). The mean graft volume (mL) in RLDRH was higher than that of ODRH, but lower than that of LADRH (696.5 SD ± 38.6; 683.4 SD ± 38.1; 785.1 SD ± 144.2, respectively). Similarly, the mean warm ischemia time in RLDRH was higher than that of ODRH (13.3 SD ± 1.6; 4.9 SD ± 1.2) (Appendix A).

#### 3.2.1. Patient Important Outcomes

##### Clavien–Dindo Complications I–II

Three studies [17,26,27] comparing RLDRH and ODRH found that the risk of complications was lower in RLDRH (RR: 0.52 95%CI 0.29, 0.94; *n* = 48; *I*^2^ = 0%). However, one study [17] compared RLDRH versus LADRH and showed no difference in the risk of complications (RR: 0.73 95%CI 0.39, 1.38; *n* = 41) (Figure 2—B).

##### Clavien–Dindo Complications III–IV

Three studies [17,26,27] comparing RLDRH and ODRH found no difference in the risk of complications (RR: 2.10 95%CI 0.44, 9.96; *n* = 6; *I*^2^ = 0%) (Figure 2—B). Likewise, one study [17] compared RLDRH versus LADRH and showed no difference in the risk of complications (RR: 2.27 95%CI 0.33, 15.67; *n* = 4) (Figure 2—B).

The complications in RLDRH were reported by these three studies (*n* = 16) and included: pleural effusion (*n* = 1, 6%), hepatic artery bleeding (*n* = 1, 6%), biliary leak (*n* = 2, 13%), and pulmonary embolism (*n* = 1, 6%) (Appendix A).

##### Mortality

One study [27] comparing RLDRH and ODRH reported no deaths in both arms (Appendix A).

##### Postoperative Pain (Visual Analogue Scale)

Two studies [17,27] comparing RLDRH and ODRH found no differences in the pain score at day > 3 (MD: −0.6 95%CI −1.6, 0.4; *n* = 219; *I*^2^ = 92%). On the contrary, one study [17] compared RLDRH versus LADRH and showed that the pain score at day > 3 was lower with RLDRH (MD: −0.4 95%CI −0.8, −0.09; *n* = 170) (Figure 3—B).

#### 3.2.2. Secondary Outcomes

##### Operative Time (Min)

Four studies [17,25,26,27] reported the mean total operative time for RLDHR (519.4 min) and ODRH (380.3 min). Only one [17] study reported the mean total operative time for LADRH (404.0 min). When comparing RLDRH and ODRH, the mean operative time was 133.4 min longer in RLDRH (95%CI 72.8, 194.1; *n* = 399; *I*^2^ = 92%). Similarly, when comparing RLDRH versus LADRH, the mean operative time was 137.7 min longer in RLDRH (95%CI 107.4, 168.0; *n* = 170) (Figure 4—A).

##### Estimated Blood Loss (mL)

Four studies [17,25,26,27] comparing RLDRH and ODRH found no difference in the estimated mean blood loss (MD: −18.2 95%CI −149.5, 113.0; *n* = 399; *I*^2^ = 94%). On the contrary, one study [17] compared RLDRH versus LADRH and showed that the total mean blood loss was lower in RLDRH (MD: −155.7 95%CI −214.6, −96.8; *n* = 3170) (Figure 4—B).

##### Postoperative Liver Function

Three studies [17,25,26] comparing RLDRH and ODRH found no differences in postoperative mean peak ALT and AST levels (MD: 11.1 95%CI −140.6, 162.8; *n* = 294; *I*^2^ = 95%; MD: −43.0 95%CI −214.4, 128.4; *n* = 294; *I*^2^ = 97%, respectively) (Figure 4—C,D), but the postoperative mean peak total bilirubin level was lower in RLDRH (MD: −0.7 95%CI −1.0, −0.4; *n* = 294; *I*^2^ = 0%) (Figure 3—A). One study [17] compared RLDRH versus LADRH and showed that the postoperative mean peak ALT and AST levels were higher in RLDRH (MD: 103.1 95%CI 68.5, 137.7; *n* = 170; MD: 71.4 95%CI 45.2, 97.6; *n* = 170, respectively) (Figure 4—C,D). Conversely, it was shown that there was no difference in the postoperative mean peak total bilirubin level (MD: −0.3 95%CI −0.7, 0.1; *n* = 170) (Figure 3—A).

##### Length of Hospital Stay (Days)

Four studies [17,25,26,27] comparing RLDRH and ODRH found that the mean LOS was lower with RLDRH (MD: −0.8 95%CI −1.4, −0.3; *n* = 399; *I*^2^ = 57%). However, one study [17] compared RLDRH versus LADRH and showed no difference in the mean LOS (MD: 0.3 95%CI −0.3, 0.9; *n* = 170) (Figure 3—C).

### 3.3. Studies Reporting Outcomes of Recipients

A total of 151 liver transplant recipients received livers after RLDRH (mean age = 57.5 SD ± 0.3; female: *n* = *33* [21.8%]), 248 after ODRH (mean age = 55.3 SD ± 6.7; female: *n* = 38 [15.3%]), and 118 after LADRH (mean age = 53.8 SD ± 11.2; female: *n* = 38 [32.2%]). The mean MELD scores in the RLDRH, ODRH, and LADRH groups were 18.1 SD ± 4.9, 17.5 SD ± 6.4, and 16.2 SD ± 8.3, respectively (Appendix A). Two studies [17,27] reported the indications for transplantation (*n* = 337). Among the three groups, hepatocellular carcinoma (*n* = 135, 40%), end-stage liver disease (*n* = 97, 29%), and nonalcoholic steatohepatitis (*n* = 44, 16%) were the most common (Appendix A).

#### 3.3.1. Patient Important Outcomes

##### Clavien–Dindo Complications I–II

Three studies [17,26,27] comparing RLDRH and ODRH found no difference in the risk of complications (RR: 1.34 95%CI 0.71, 2.55; *n* = 37; *I*^2^ = 0%). One study [17] comparing RLDRH versus LADRH showed similar results (RR: 1.00 95%CI 0.53, 1.87; *n* = 36) (Figure 2—C).

##### Clavien–Dindo Complications III–IV

Three studies [17,26,27] comparing RLDRH and ODRH found no difference in the risk of complications (RR: 0.71 95%CI 0.43, 1.16; *n* = 92; *I*^2^ = 34.9%). Similar results were found when RLDRH was compared to LADRH [17] (RR: 0.66 95%CI 0.39, 1.11; *n* = 58) (Figure 2—C).

The most common complications in RLDRH, reported by three studies, were (*n* = 43): biliary leak (*n* = 12, 28%), and hepatic artery thrombosis (*n* = 5, 12%) (Appendix A). As for the conversion rate, there was only one conversion from robotic to open [17]. Minilaparotomy was performed due to injury to the S2 bile duct. The authors reported that the cause of the conversion was not related to the robotic techniques, but to the rare occurrence of an anomaly and lack of indocyanine green cholangiogram guidance [17].

##### Mortality

There was no difference in the risk of mortality in three studies [17,25,27] comparing RLDRH and ODRH (RR: 1.20 95%CI 0.55, 2.62; *n* = 23; *I*^2^ = 0.0%) and in one study [17] that compared RLDRH and LADRH (RR: 2.27 95%CI 0.14, 35.59; *n* = 2) (Figure 2—A).

#### 3.3.2. Secondary Outcomes

##### Estimated Blood Loss (mL)

One study [27] comparing RLDRH and ODRH found no difference in the estimated mean blood loss (MD: 500.0 95%CI −1387.3, 2387.3; *n* = 105) (Figure 4—E).

##### Postoperative Liver Function

Postoperative mean peak ALT and AST levels and mean total bilirubin levels were reported in one study [26]. This study compared RLDRH and ODRH and showed that there were no differences in postoperative mean peak ALT and mean total bilirubin levels (MD: 9.2 95%CI −23.3, 41.7; *n* = 67; MD: 0.0 95%CI −0.0, 0.0; *n* = 67, respectively) (Figure 4—F and Figure 3—E). On the contrary, this study found that the postoperative mean peak AST level was lower with RLDRH (MD: −0.5 95%CI −0.9, −0.1; *n* = 67) (Figure 3—D).

##### Length of Hospital Stay (Days)

Two studies [17,27] comparing RLDRH and ODRH found no significant difference in the mean LOS (MD: −0.7 95%CI −5.7, 4.3; *n* = 219; *I*^2^ = 0%). Conversely, one study [17] compared RLDRH versus LADRH and showed that RLDRH was significantly linked to a reduced mean LOS compared to LADRH (MD: −6.4 95%CI −11.3, −1.5; *n* = 170) (Figure 3—F).

## 4. Discussion

The present meta-analysis showed that robotic donor hepatectomy seems to be a feasible alternative for open donor hepatectomy in living liver donation. Although its main drawback seems to be prolonged operative times, its equal or even lower risk of postoperative complications and similar post-hepatectomy liver function and pain outcomes render this approach potentially useful and safe for LDLT. Finally, differences were not found in recipient outcomes after RLDRH when compared to other approaches.

There are emerging data demonstrating a clear role for minimally invasive approaches in the modern era of hepatobiliary and pancreatic surgery [28,29,30,31,32,33,34]. The implementation of this technology in the field of liver transplantation showed that although laparoscopic living donor hepatectomy is theoretically challenging due to complex vascular and biliary variations, it is shown to be an effective alternative to the open approach [35,36]. Moreover, it offers significant reductions in donor cosmetics and trauma, which still remain a major concern among potential adult living donors [37,38,39,40].

However, despite the initial enthusiasm around the technical feasibility of the minimally invasive living liver donation, there are still concerns for its widespread application, especially in Western countries, since the mortality risk was estimated to range between 0.2% and 1% and complication rates varied from 20% to 40% [41,42]. Nonetheless, a recent meta-analysis showed that these outcomes are not inferior to the open approach [8,43].

Robotic liver resections are becoming popular due to the advantages of robotic surgical systems over laparoscopy. These advantages mainly consist of the availability of endo-wristed instruments and a 3D view, which can be really useful in cases of meticulous dissection of the liver hilum and safe dissection of the inferior vena cava and hepatic veins. Many retrospective studies and meta-analyses showed that the outcomes of robotic major hepatectomy are not inferior overall, when compared with open or laparoscopic approaches, with the only shortcomings being the operative time and associated cost [44,45,46,47].

In the setting of liver transplantation, most of the published data on robotic living donor hepatectomy come from centers with extensive experience, unique skillsets, and high volumes of robotic cases that do not facilitate the generalizability of the results. Most available studies had set strict initial selection criteria for both donors and recipients to combine the goals of non-inferior transplant-related outcomes and advantages of minimally invasive surgery (cosmetics, less pain, shorter LOS). Some experts also recommend that the initial indications for LDLT for a right graft should be a graft-to-recipient weight ratio >1.0, remnant liver volume > 35%, normal vascular and biliary anatomy, and a nonemergent setting [8,47]. Still, very few donors outside these criteria were included in the published studies.

The present study has several strengths and limitations that must be mentioned. Firstly, to our knowledge, this is the first meta-analysis on the outcomes of robotic living donor hepatectomy focusing on both liver donors and liver transplant recipients. Moreover, the search of the literature was comprehensive, following a systematic methodology by applying pre-specified and detailed data tabulation and extraction, and standardized evaluation of evidence quality and publication bias. All steps were performed rigorously by multiple researchers. This approach facilitated the identification of a “clean” dataset from comparative studies of different approaches to allow better generalizability of the results.

Nonetheless, the present study has certain limitations. Firstly, as all of the included studies were comparative cohort studies conducted in the Asian continent, this meta-analysis might be prone to selection bias inherent to the included studies, and the results may not be applicable in other countries. Moreover, we included a conference abstract that may increase the risk of bias. Additionally, some of the eligible studies lacked data granularity on all characteristics or outcomes of interest, and thus the relative rates were estimated accordingly based on the availability of data. Additionally, the majority of studies did not specify how the decision was made to proceed with each surgical approach. As all included patients underwent right hepatectomy, the two comparison arms were generally homogeneous, and in each study, all resections were performed by an experienced team of minimally invasive liver surgeons, both of which minimize the risk of confounding variables.

Overall, our opinion is that, despite being in its infancy, robotic surgery is here to stay. These early results are really encouraging. Additionally, robotic surgery can be a solution to the well-known technical difficulties of laparoscopy mainly related to suboptimal instrumentation and challenging ergonomics as well as the demanding learning curve, especially as far as the pure laparoscopic approach is concerned. Despite being counterintuitive, robotic surgery has a lot of similarities with open surgery and practically emulates the steps of open surgery in a closed abdomen with the addition of the benefits of minimally invasive surgery. An increasing number of transplant centers are introducing robotic surgery in their clinical practice, and our anticipation is that robotic surgery, mainly in donor surgery, will be part of the current practice in the coming years.

## 5. Conclusions

Low- to unclear-quality evidence shows that robotic donor hepatectomy might be a safe approach for living liver donation compared to LADRH and ODRH. The results of this study should be interpreted with caution as the number of participants was low and all the included studies comprised only Asian patients. However, there is a current trend and associated benefits of using robotics, which should not be overlooked. Further experimental studies are needed to confirm these results and draw robust conclusions.

## Figures and Tables

**Figure 1 jcm-11-02603-f001:**
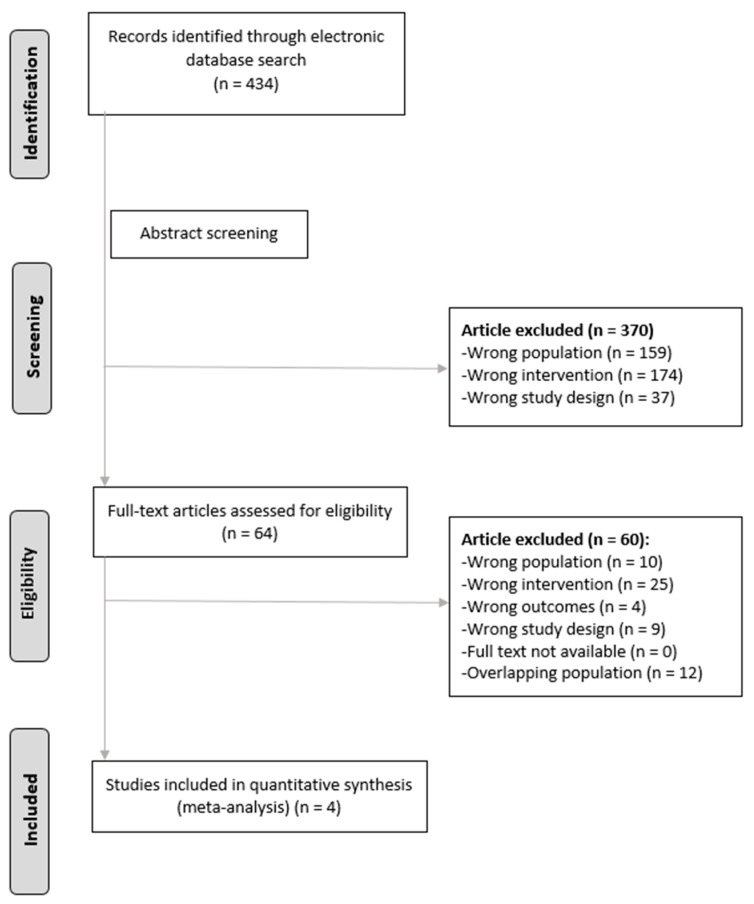
Preferred Reporting Items for Systematic Reviews and Meta-Analyses (PRISMA) flow diagram of the study selection process.

**Figure 2 jcm-11-02603-f002:**
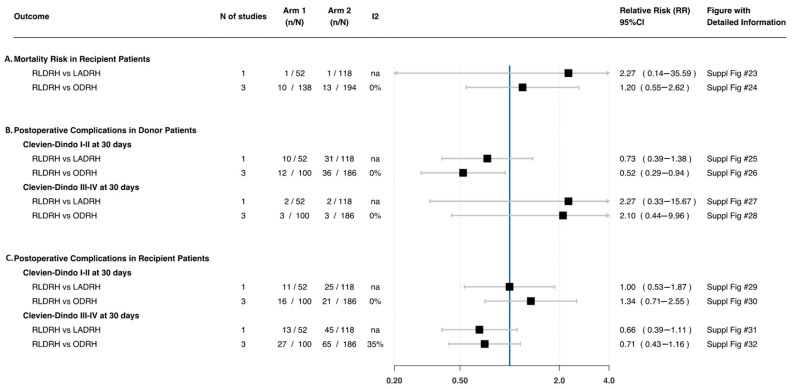
Risk ratio of postoperative Clavien–Dindo complications and mortality in recipients and donors. Suppl figs can be found in Github, please refer to Section 2.8.

**Figure 3 jcm-11-02603-f003:**
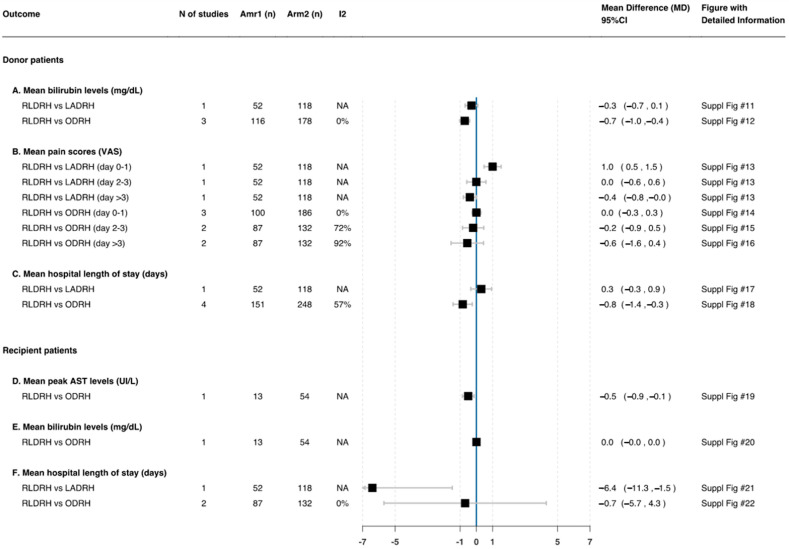
Mean differences in donors’ and recipients’ postoperative characteristics—part 1. Suppl figs can be found in Github, please refer to Section 2.8.

**Figure 4 jcm-11-02603-f004:**
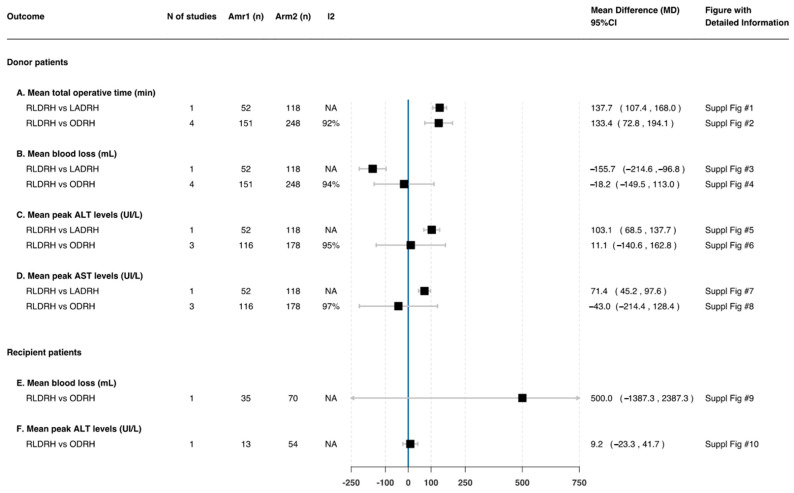
Mean difference in donors’ and recipients’ postoperative characteristics—part 2. Suppl figs can be found in Github, please refer to Section 2.8.

**Table 1 jcm-11-02603-t001:** General characteristics of included studies.

Authors/Year	Country	Study Design	Conference Abstracts	Study Period	Setting	Sample Size (*n*)	Robotic (*n*)	Laparoscopic (*n*)	Open (*n*)
Rho et al., 2020	South Korea	Retrospective	NA	March 2016 to March 2019	Single center	232	52	62	118
Broering et al., 2020	Saudi Arabia	Retrospective	NA	January 2015 to July 2019	Single center	105	35	NA	70
Binoj et al., 2020	India	Prospective	American Association for the Study of Liver Diseases Congress 2020	NR	NR	113	51	NA	62
Chen et al., 2016	China	Retrospective	NA	June 2005 to September 2012	Single center	67	13	NA	54

NR: not reported, NA: not applicable.

## Data Availability

The data were extracted from already published studies and thus can be found publicly available in the respective articles. All documents used in the screening phases, tables, figures, and analysis codes are freely available at Github: https://github.com/ponceoscarj/Hepatectomy/blob/main/hepatectomy.md (accessed on 23 March 2022).

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
