# Peer review of "Robotic Living Donor Right Hepatectomy: A Systematic Review and Meta-Analysis"

_jcm, 2022, doi:10.3390/jcm11092603_

Round 1

Reviewer 1 Report

I read with interest the paper and the meta analysis on robotic living liver donation. 

Results are clear, however, it would be interesting to have a more critical and personal view of the robotic approach by the authors in light of these results. For example, considering the almost equal post-operative pain and the slight advantage in the length of stay, you consider the robotic approach worthwhile? it is also interesting to see that the blood loss is almost equal among the groups. 

Also, it would be interesting to have the patient selection criteria for the robotic approach, to see if it is similar among studies. 

Also, a recent meta-analysis has been published on the mini-invasive approach that could be part of the discussion 10.1007/s13304-021-01160-x

Author Response

We appreciate the reviewer’s comments and suggestions. We added our personal view on the potential role of robotic surgery in liver transplantation.

We added the patient criteria for the robotic approach in the results section under 3.2. You can also find it in Supplemental Digital Content Table S1. It reads as follow:

“Only one study reported patients’ selection criteria and stated that candidates were at will to choose the robotic procedure after passing inclusion criteria (Supplemental Table S1)”

Reviewer 2 Report

In this systematic review and meta-analysis, the authors find that robotic living donor right hepatectomy is a safe and feasible option. As acknowledged by them, the included studies are from high volume experienced centers and the quality of evidence is low to support uniform adoption of robotic donor hepatectomy as a standard of care.

It can be highlighted that Grade III/IV (major) complications in donors (including bile leak and bleeding) undergoing robotic right living donor hepatectomy was not higher as compared to open/laparoscopic approaches. From the available data, It also seems like recipient outcomes are not different.

Details of any significant intra-operative events or reasons for conversion to open approach if any available for individual studies can be discussed

Warm ischemia time (time from clamping of right hepatic artery to placement of graft in cold preservative solution for bench perfusion) may be an important factor that can be compared across robotic, laparoscopic and open arms 

Author Response

We appreciate the reviewer is bringing these important points. We agree and added into the result and discussion section.

Warm ischemia time information can be find under section 3.2, supplemental Digital Content Table S2, and in the discussion section. It reads as follow:

“Similarly for the mean warm ischemia time in RLDRH, which was higher than that of ODRH (13.3 SD±1.6; 4.9 SD±1.2) (Supplemental Table S2)”

Details of any significant intra-operative events or reasons for conversion to open approach can be find under section 3.3.1.2 and in the discussion section. It reads as follow:

“As far for the conversion rate, there was only 1 conversion from robotic to open [17]. Minilaparotomy was performed due to injury to the S2 bile duct. The authors reported that the cause of the conversion was not related to the robotic techniques, but to the rare occurrence of an anomaly and lack of indocyanine green cholangiogram guidance [17]”

Reviewer 3 Report

This paper represents a remarkable achievement by the authors and is of interest in the field. A few points need to be addressed.

In study selection, it was described only that the reviewers have decided on the eligibility for inclusion/exclusion for the systematic review. Please address a detailed study selection for eligibility for systematic review.

Please address the detailed reasons for the exclusion of 370 articles/studies during screening.

Please add a table presenting studies regarding robotic living donor right hepatectomy, including the excluded studies, describing patient/case number, study period, study setting etc. It would be helpful for readers to understand the flow of systematic reviews and a recent trend of study.

Did the authors try to collect undescribed data for studies excluded from the systematic review? Four studies seem to be insufficient for systematic meta-analysis, even considering that robotic donor right hepatectomy is not widely performed.

Author Response

We thank the reviewer for the comments. We have updated the eligibility for inclusion/exclusion for the systematic review. It can be find under section 2.1 and reads as follow:

“Observational or experimental comparative studies assessing the safety of RLDRH in patients older than 18 years who require transplantation, regardless of the underlying liver disease, compared to ODRH or LADRH, were included. Patients with normal anatomy or multiple bile ducts and portal trifurcation, remnant liver more than 30%, and graft-to-recipient weight ratio over 0.8% were included.”

We have also updated the flow chart and included the reasons for exclusion in the abstract screening phase.

We agree that describing patient/case number, study period, study setting etc. of studies regarding RLDRH is helpful. We already had that information in Table 1. However, although understanding the recent trend of studies is important, we disagree that describing characteristics of excluded studies in this systematic review is helpful. This is because 90% of excluded studies in the abstract screening and 60% of excluded studies in the full-text screening were excluded due to wrong population/intervention. In other words, these excluded articles were not related to liver transplantation or robotic surgery.

We agree with your comment. We have added in the section 2.5 author contact that besides contacting authors of the included studies, we had also contacted authors of excluded articles due to wrong outcomes. However, only one author replied. It reads as follows:

“We emailed the first and corresponding authors of the included studies and excluded articles due to wrong outcomes, to request further information”

Round 2

Reviewer 3 Report

In Supplemental Digital Content Table S1 of Rho et al, “inferior high hepatic vein” is supposed to be “inferior right hepatic vein”. Additionally, several typos (Receptor instead of recipient, min instead of mean etc) are noted in Supplemental Digital Content Tables. Please correct the spelling.

“Robotic/laparoscopic/open transplantations” described in Supplemental Digital Content Tables have the possibility of misunderstanding. Donor hepatectomy seems to be more appropriate instead of transplantation.  

Author Response

Thank you for your comments. We addressed all typos as requested